# Salivary IgA and IgG Antibody Responses against Periodontitis-Associated Bacteria in Crohn’s Disease

**DOI:** 10.3390/ijms24032385

**Published:** 2023-01-25

**Authors:** Mervi Gürsoy, Jaana Rautava, Pirkko Pussinen, Anna Karin Kristoffersen, Morten Enersen, Vuokko Loimaranta, Ulvi Kahraman Gürsoy

**Affiliations:** 1Department of Periodontology, Institute of Dentistry, University of Turku, 20014 Turku, Finland; 2Welfare Division, Oral Health Care, 20540 Turku, Finland; 3Department of Oral and Maxillofacial Diseases, Clinicum, Faculty of Medicine, University of Helsinki and Helsinki University Hospital, 00014 Helsinki, Finland; 4Department of Pathology, Medicum, Faculty of Medicine, University of Helsinki and HUS Diagnostic Center, HUSLAB, Helsinki University Hospital, 00260 Helsinki, Finland; 5Oral and Maxillofacial Diseases, University of Helsinki, 00014 Helsinki, Finland; 6Institute of Dentistry, University of Eastern Finland, 70210 Kuopio, Finland; 7Institute of Oral Biology, Faculty of Dentistry, University of Oslo, 0372 Oslo, Norway

**Keywords:** Crohn’s disease, immune response, plasma cells

## Abstract

Elevated serum immunoglobulin (Ig) antibody levels are observed in Crohn’s disease patients. The aim of this study was to evaluate the salivary IgA and IgG antibody levels against *Porphyromonas gingivalis*, *Tannerella forsythia*, *Aggregatibacter actinomycetemcomitans*, and *Prevotella intermedia* in Crohn’s disease patients. Eighty-eight participants (47 Crohn’s disease patients and 41 systemically healthy age- and gender-matched controls) were included in the study. Oral and medical health statuses were recorded and salivary samples were collected. Salivary *P. gingivalis*, *T. forsythia*, *A. actinomycetemcomitans*, and *P. intermedia* carriage were analyzed with DNA sequencing technique, salivary levels of IgG1, IgG2, IgG3, IgG4, and IgM were measured with the Luminex^®^ xMAP™ technique, and salivary IgA and IgG antibody levels against *P. gingivalis*, *T. forsythia*, *A. actinomycetemcomitans*, and *P. intermedia* were detected by ELISA. As result, higher salivary IgG2 (*p* = 0.011) and IgG3 (*p* = 0.006), *P. gingivalis* IgA (*p* < 0.001), *A. actinomycetemcomitans* IgG (*p* = 0.001), and *P. intermedia* IgG (*p* < 0.001) antibody levels were detected in the Crohn’s disease group compared to the controls. Salivary *P. gingivalis* carriage was lower in the Crohn’s disease group in comparison to the controls (*p* = 0.024). In conclusion, salivary IgA antibody responses against *P. gingivalis* and IgG antibody responses against *P. intermedia* have independent associations with Crohn’s disease.

## 1. Introduction

Crohn’s disease is an inflammatory bowel disease with chronic and relapsing-remitting character. Crohn’s disease has multifactorial etiology, and the cellular and humoral immune responses play significant roles in their pathogenesis [1]. Crohn’s disease can be observed at any part of the gastrointestinal tract, including the oral cavity [2].

Early studies identified three common immune response disruptions in Crohn’s disease patients; excessive T-cell activation, decreased T-cell suppression, and elevated B-cell activation [3]. In line with these findings, recent studies also demonstrated increased mucosal cells expression of interleukin (IL)-23/tumor necrosis factor (TNF)-α and activated Th17/Th1 cascade in Crohn’s disease [2]. Yet, the interest in the contribution of B cells and their immunoglobulins to Crohn’s disease pathogenesis accelerated recently. Secretory immunoglobulins maintain the mutual symbiotic relationship between microorganisms and the host. An elevated number of immunoglobulin (Ig) secreting cells [3] and an increased number of IgA- and IgG-bound commensal bacteria [4] were observed in Crohn’s disease patients. Indeed, increased serum IgA and IgG levels in Crohn’s disease were related to disease severity [5]. Yet, the elevated immunoglobulin levels in Crohn’s disease cannot be fully explained with the gastrointestinal mucosal barrier dysfunctions or acute inflammatory flares [6].

Shared similarities in the ethiopathogenesis of periodontitis and Crohn’s disease include defective neutrophil chemotaxis, impaired TNF-α expression, and atypical Th1 cell response [7]. Moreover, the intraoral abundance of some periodontitis-associated bacteria, genus *Prevotella* and *Veillonella*, increase in Crohn’s disease patients [8]. In the present study, we hypothesized that the shifts in immunoglobulin response in Crohn’s disease patients reflect in salivary levels of species-specific IgA and IgG antibody levels. Therefore, the aim of this study was to analyze the salivary levels of IgA and IgG antibodies against *Porphyromonas gingivalis*, *Tannerella forsythia*, *Aggregatibacter actinomycetemcomitans*, and *Prevotella intermedia* in Crohn’s disease patients.

## 2. Results

Detailed descriptions of the demographic data were published previously [9]. Briefly, the percentage of males was 23.4 % in the Crohn’s disease group and 29.3% in the control group (*p* = 0.629), the mean age was (46.4 ± 13.9) in the Crohn’s disease group and (47.4 ± 13.2) in the control group (*p* = 0.734), and the percentage of the periodontally healthy individuals was 25% in the Crohn’s disease group and 36.6% in the control group (*p* = 0.021). No difference was observed in stimulated salivary flow levels between Crohn’s disease (mean 1.42 ± 0.72 mL/min) and control (1.59 ± 0.9 mL/min) groups (*p* = 0.313). According to the short CDAI score, 53.2% of the Crohn’s disease patients were in remission (short CDAI < 150 points), 31.9% had mild disease (short CDAI 150–219 points), 14.9% had moderate disease (short CDAI 220–450 points), and none had severe disease (CDAI > 450 points).

Salivary carriage of *P. gingivalis*, *T. forsythia*, *A. actinomycetemcomitans*, and *P. intermedia* are given in Table 1. A higher percentage of individuals with *P. gingivalis* positivity was observed in the control group in comparison to Crohn’s disease group (*p* = 0.040). No other significant difference was observed in the salivary carriage of *T. forsythia*, *A. actinomycetemcomitans*, and *P. intermedia.*

Salivary levels of total IgA, IgG1, IgG2, IgG3, IgG4, and IgM antibody levels are given in Table 2. Salivary IgG2 (*p* = 0.011) and IgG3 (*p* = 0.006) antibody levels were found to be higher in Crohn’s disease group in comparison to the controls.

Salivary IgA and IgG antibody levels against *P. gingivalis, T. forsythia, A. actinomycetemcomitans*, and *P. intermedia* are given in Table 3. Salivary *P. gingivalis* IgA (*p* < 0.001), *A. actinomycetemcomitans* IgG (*p* = 0.001) and *P. intermedia* IgG (*p* < 0.001) antibody levels were higher in the Crohn’s disease group compared to the controls.

Linear regression models indicated independent associations of elevated salivary *P. gingivalis* IgA and *P. intermedia* IgA levels with Crohn’s disease, after adjustments with periodontitis diagnosis and carriage of salivary bacteria (Table 4).

## 3. Discussion

To our knowledge, this is the first study to evaluate the salivary levels of specific IgA and IgG antibodies against *P. gingivalis, T. forsythia, A. actinomycetemcomitans*, and *P. intermedia* in patients with Crohn’s diseases. According to our findings, elevated salivary IgA response against *P. gingivalis* and *P. intermedia* are independently associated with Crohn’s disease, and these associations are not related to the salivary levels of these bacteria.

The current study participants were recruited from the Crohn and Colitis patient organization (IBD Association of Finland) during January 2017 to May 2018. All Crohn’s disease patients of the association received the invitation and those who were willing to participate and fit to the inclusion criteria were recruited. Therefore, no sample size calculation was performed. The main strength of the present study was the comprehensive evaluation of bacteria-specific IgA and IgG antibody responses together with the salivary levels of bacteria and secretory immunoglobulins. With this, it was possible to argue on the possible explanations of the elevated immunoglobulin antibody responses. One limitation of the study was that the majority of the Crohn’s disease patients (89.4%) were under medication for Crohn’s disease and 34.0% had undergone surgical treatment as part of their Crohn’s disease treatment. Use of immunosuppressive drugs may hinder the real effect of the Crohn’s disease on B-cell functions. The localization of the lesions and disease behavior according to Montreal classification were not available, therefore we could not relate these parameters with the salivary findings. Finally, the lack of serum Ig antibody levels did not allow us to evaluate the systemic levels of these Ig antibodies and compare those with their levels in saliva.

According to the present results, salivary IgG2, IgG3, and IgM antibody levels were higher in Crohn’s disease patients than in controls. Effector functions of Ig antibodies are regulated by their Fc regions. IgM, IgG2, and IgG3 activate the complement system, while IgA takes part in mucosal immunity. Previous studies showed increased serum IgG2 levels in Crohn’s disease patients and increased serum IgG1 levels in patients with ulcerative colitis, which is another form of inflammatory bowel disease, indicating a distinct regulation of IgG subclasses among these two diseases [10,11]. The IgG2 antibody response is not always T-cell dependent, but various bacterial polysaccharides may also evoke this response [12]. Therefore, the observed increase in salivary IgG2 levels in Crohn’s disease can be an outcome of shift in microbiome composition in this specific patient group. Increased levels of salivary IgA antibodies were demonstrated in Crohn’s disease patients [13]. This finding was explained with the frequent observation of aphthous stomatitis in patients with Crohn’s disease, as salivary IgA1 and IgA2 antibodies increase during flare-up phases of recurrent [14]. In the present study, Crohn’s disease-associated mucosal lesions were diagnosed only in five individuals, which may suggest that the aphthous ulcers cannot fully explain the increase in salivary Ig antibody levels.

Specific IgG and IgG antibody responses against antigens of some specific microorganisms was previously demonstrated in patients with Crohn’s disease. Among these antigens, the most studied ones were oligomannose epitope on the *Saccharomyces cerevisiae* [6], outer membrane porin C of *Escherichia coli* [15], and some bacterial flagella [16]. According to our results, salivary *P. gingivalis* IgA, *A. actinomycetemcomitans* IgG and *P. intermedia* IgG antibody levels were higher in Crohn’s disease group compared to the controls. *P. gingivalis*-specific IgG and IgA antibodies in serum were related to prognosis of esophageal squamous cell carcinoma [17] and rheumatoid arthritis [18]. Children with juvenile idiopathic arthritis also have higher IgG antibody responses against *P. gingivalis* and *P. intermedia* [19]. Moreover, serum IgG antibody responses against *P. gingivalis* and *A. actinomycetemcomitans* were associated with the formation and rupture of intracranial aneurysms [20]. According to our findings, the abundance of *P. gingivalis* or *P. intermedia* in the oral cavity do not interrupt the associations of Crohn’s disease with salivary IgA antibody responses against *P. gingivalis* and IgG antibody responses against *P. intermedia.* Yet, as the present study is the first to demonstrate the salivary Ig antibody responses against periodontitis-associated bacteria, it was not possible for us to discuss the contribution of the aggravated systemic immunological response against these bacteria.

In conclusion, salivary IgA antibody responses against *P. gingivalis* and IgG antibody responses against *P. intermedia* are associated with Crohn’s disease, and this association is independent of the salivary abundances of *P. gingivalis* and *P. intermedia*.

## 4. Materials and Methods

### 4.1. Study Design and Population

Altogether, 88 individuals (47 Crohn’s disease patients and 41 systemically healthy age- and gender-matched controls) participated in the study. Exclusion criteria were being diagnosed with diabetes mellitus, having periodontitis as a manifestation of systemic disease, smoking, being pregnant or lactating, excessive use of alcohol, having less than 24 teeth, and using antimicrobial medication in the preceding six months prior to the clinical examination. Patient recruitment, clinical evaluations, and saliva collection took part between January 2017 and May 2018. The Ethical Committee of the Hospital District of Southwest Finland approved the study protocols (114/1801/2016), and the study was conducted according to the guidelines of the World Medical Association Declaration of Helsinki. Participants received written information on study protocols and purposes, and all participants gave their written consent.

### 4.2. Clinical Examinations

Information on general health, medications, and previous dental care, and the demographic data were collected by structured questionnaires. The full-mouth clinical examinations (oral mucosal, cariological, and periodontal) were performed by a specialist in oral pathology (JR) and by a periodontist (MG). Detailed information on clinical examinations can be found elsewhere [9].

### 4.3. Saliva Collection

The study participants were instructed to avoid eating and brushing their teeth 30 mins prior to the visit. Five ml of paraffin-stimulated saliva samples were collected from each patient before the clinical examinations. All samples were stored at −70 °C until their further use.

### 4.4. Determination of Salivary Immunoglobulin A (IgA), G (IgG), and M (IgM) Antibody Levels

Saliva samples were centrifuged at 10,000 rpm for 5 min, and the supernatants were used to detect salivary immunoglobulin antibody levels. Salivary levels of IgG1, IgG2, IgG3, IgG4, and IgM were measured with the Luminex^®^ xMAP™ technique (Luminex Corporation, Austin, TX, USA) using commercially available kits (Bio-Plex Pro™ Human Isotyping Panel, Bio-Rad, Santa Rosa, CA, USA).

Salivary levels of total IgA were measured by trapping-antibody-type ELISA assay as described earlier [21], with some modifications. Briefly, polystyrene microtiter plates were coated with anti-human IgA antibodies (dil 1:1000, Dako, Glostrup, Denmark) over night at +4 °C. The plates were blocked with 1% BSA and saliva, (dil 1:1000) was added. After two hours of incubation and washing, anti-human IgA-HRP conjugate (dil 1:5000, Dako, Glostrup, Denmark) was added for one hour. After three washes, o-phenylenediamine was added and the absorbance (492 nm) was measured. Human serum protein calibrator (Dako, Glostrup, Denmark) was used to create a standard curve and calculate the amount of IgA in saliva samples. All samples were analyzed in triplicates

### 4.5. Determination of Salivary IgA and IgG Antibody Levels against Porphyromonas gingivalis, Tannerella forsythia, Aggregatibacter actinomycetemcomitans, and Prevotella intermedia

Saliva samples were centrifuged at 10,000 rpm for 5 min, and the supernatants were used to detect salivary immunoglobulin antibody levels. The methods have been published earlier in detail [22,23]. The antigens were composed of killed whole-cell preparations of each species. The saliva samples were diluted (1:10), and the antibody levels were measured in duplicates. The mean absorbance values were normalized per reference sample on each plate and expressed as continuous absorbance units (AU).

### 4.6. Determination of Salivary Carriage of P. gingivalis, T. forsythia, A. actinomycetemcomitans, and P. intermedia

The detection of salivary microbial abundance was a part of a larger microbiome project. Briefly, Epicentre Masterpure complete DNA extraction kit (MC85200 Lucigen, Biosearch Technologies, Middleton, WI, USA) was used to isolate bacterial DNA from the saliva samples. The primers (341F/805R) were designed to target the V3 and V4 regions of 16S rRNA. For sequencing on the NovaSeq platform paired-end reads (2 × 250 bp), a PCR library was constructed. After the completion of NGS sequencing, the raw data were obtained. High-quality clean data were obtained after merging pair-end with overlap, quality control, and Chimera filtering. Data are presented as salivary bacterial carriage.

### 4.7. Statistical Analyses

The SPSS statistical program (version 26.0; IBM Corp., Armonk, NY, USA) was used in data analyses. The data distributions of salivary immunoglobulin levels were skewed, therefore non-parametric Mann–Whitney U tests were applied. In comparison of salivary carriage of bacteria, the Chi-Square test was used. A linear regression analysis was performed to study the association of salivary Ig antibody levels with Crohn’s disease by controlling for periodontitis diagnosis and carriage of salivary bacteria tested.

## Figures and Tables

**Table 1 ijms-24-02385-t001:** Salivary carriage of *P. gingivalis* (Pg), *T. forsythia* (Tf), *A. actinomycetemcomitans* (Aa), or *P. intermedia* (Pi) in Crohn’s disease and control groups.

A)	*Pg* Prevalence (%)	*Tf* Prevalence (%)	*Aa* Prevalence (%)	*Pi* Prevalence (%)
Crohn’s disease	4.3%	31.9%	0%	6.4%
Controls	19.5%	39.0%	7.3%	7.3%
*p* value	0.040	0.510	0.097	1.00

*p*-values derived from Chi-Square tests.

**Table 2 ijms-24-02385-t002:** Salivary levels of IgA, IgG1, IgG2, IgG3, IgG4, and IgM antibodies in Crohn’s disease and control groups.

	IgA (µg/mL)	IgG1 (ng/mL)	IgG2 (ng/mL)	IgG3 (ng/mL)	IgG4 (ng/mL)	IgM (ng/mL)
Crohn’s disease	36.6 (21.6)	42.1 (64.7)	261 (286)	34.1 (46.3)	9.13 (11.2)	40.2 (72.9)
Controls	31.6 (31.6)	34.5 (66.6)	169 (208)	20.4 (28.4)	8.35 (15.1)	52.6 (112)
*p* value	0.074	0.197	0.011	0.006	0.631	0.099

Data are presented as medians and interquartile ranges (in parenthesis). *p*-values derive from Mann–Whitney U test.

**Table 3 ijms-24-02385-t003:** Salivary IgA and IgG antibody levels against *P. gingivalis (Pg), T. forsythia (Tf), A. actinomycetemcomitans (Aa)*, and *P. intermedia (Pi)* in Crohn’s disease and control groups.

	*Pg*		*Tf*		*Aa*		*Pi*	
	IgA	IgG	IgA	IgG	IgA	IgG	IgA	IgG
	AU median (IQR)							
Crohn’s disease	0.39 (0.14)	0.14 (0.04)	0.31 (0.17)	0.15 (0.06)	0.26 (0.08)	0.12 (0.02)	0.22 (0.07)	0.11 (0.02)
Controls	0.23 (0.13)	0.14 (0.06)	0.36 (0.20)	0.17 (0.07)	0.25 (0.08)	0.09 (0.03)	0.22 (0.09)	0.09 (0.02)
*p* value	<0.001	0.190	0.135	0.078	0.298	0.001	0.651	<0.001

**Table 4 ijms-24-02385-t004:** Associations of *P. gingivalis* (Pg) IgA, *A. actinomycetemcomitans* (Aa) IgG, and *P. intermedia* (Pi) IgG with Crohn’s disease. Systemically healthy group was taken as the reference.

Dependent Variable	Independent Variable	Unadjusted	Model 1	Model 2
Pg IgA	Crohn’s disease	B: 0.137, β:0.454 (95% CI: 0.080–0.195),	B: 0.138, β:0.458 (95% CI: 0.080–0.196),	B: 0.143, β:0.473 (95% CI: 0.084–0.202),
		*p* < 0.001	*p* < 0.001	*p* < 0.001
Aa IgG	Crohn’s disease	B: −0.010,	B: −0.012,	B: 0.011, β: 0.063
		β: −0.054 (95% CI: −0.049–0.029),	β: −0.068 (95% CI: −0.05–0.025),	(95% CI: −0.020–0.042),
		*p*: 0.620	*p*: 0.516	*p*: 0.469
Pi IgG	Crohn’s disease	B: 0.022,	B: 0.021,	B: 0.021,
		β: 0.496 (95% CI: 0.013–0.030),	β: 0.490 (95% CI: 0.013–0.029),	β: 0.486 (95% CI: 0.013–0.029),
		*p* < 0.001	*p* < 0.001	*p* < 0.001

Data are presented as Unstandardized B, Standardized β coefficient, 95% Confidence Interval for B (in parenthesis), and *p*-values. Model 1: adjusted for periodontitis diagnosis; Model 2: adjusted for periodontitis diagnosis and carriage of salivary bacteria (prevalence of *P. gingivalis* for Pg IgA, *A. actinomycetemcomitans* for Aa IgG, and *P. intermedia* for Pi IgG).

## Data Availability

The datasets for this study are available on request.

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
