# Peer review of "Salivary IgA and IgG Antibody Responses against Periodontitis-Associated Bacteria in Crohn’s Disease"

_ijms, 2023, doi:10.3390/ijms24032385_

Round 1

Reviewer 1 Report

In the present case-control study, Gursoy et al evaluated some periodontitis-associated bacteria and relative immunoglobulins in the saliva of 41 patients with Crohn’s disease, compared to controls. Main comments:

1) Authors did not report some disease associated characteristics, such as localization and behaviour according to Montreal classification and disease activity. For example, how many CD had upper gastrointestinal (L4) involvement? This could correlate with some salivary findings.

2) Sample size calculation is missing.

3) The way of reporting multivariate analysis is odd. Beta coefficient confidence intervals are necessary. What do models 1-2 include? Which are the other factors entered in the analysis and what is their beta?

4) Did the patients have to bush teeth before collecting saliva?

5) Was aphtous stomatitis observed in enrolled patients?

6) Page 5 lines 166-170: how can the Authors justify such finding?

Author Response

In the present case-control study, Gursoy et al evaluated some periodontitis-associated bacteria and relative immunoglobulins in the saliva of 41 patients with Crohn’s disease, compared to controls.

Authors’ response: Authors thank to the reviewer for the constructive comments

Main comments:

1) Authors did not report some disease associated characteristics, such as localization and behaviour according to Montreal classification and disease activity. For example, how many CD had upper gastrointestinal (L4) involvement? This could correlate with some salivary findings.

Authors’ response: In the present study short CDAI index (Thia et al., Inflamm Bowel Dis. 2011) was used, which classifies the Crohn’s disease activity. We agree with the reviewer that the localization and the extent of the disease may have a relation with the salivary immunoglobulin levels, however, this information was not available.

In the revised manuscript, outcomes of short CDAI are given and the lack of information on the localization of Crohn’s disease lesions are discussed in discussion under limitations (page 4 lines 142-145 and page 5 lines 146-148).

2) Sample size calculation is missing.

Authors’ response: The current study participants were recruited from the Crohn and Colitis patient organization (IBD Association of Finland) during January 2017 to May 2018. All Crohn’s disease patients of the association were received the invitation and those, who were willing to participate and fit to the inclusion criteria were recruited. Therefore no sample size calculation was performed. This information is given in discussion (page 4 lines 131-135).

3) The way of reporting multivariate analysis is odd. Beta coefficient confidence intervals are necessary. What do models 1-2 include? Which are the other factors entered in the analysis and what is their beta?

Authors’ response: We apologize for the missing footnote. Table 4 is improved as suggested.

4) Did the patients have to bush teeth before collecting saliva?

Authors’ response: The study participants were instructed to avoid tooth brushing and eating 30 min prior to the visit. This information is implemented in the text (page 6 lines 208-209)

5) Was aphtous stomatitis observed in enrolled patients?

Authors’ response: During the clinical examinations, an experienced oral pathologist evaluated the oral mucosal tissues for potential lesions, including sulcular ulceration, aphthous stomatitis, mucosal cobblestone appearance, lip swelling, angular cheilitis, mucogingivitis. No aphthous stomatitis were diagnosed.

6) Page 5 lines 166-170: how can the Authors justify such finding?

Authors’ response: The sentence is rephrased (page 5 lines 178-180)

Reviewer 2 Report

In their paper entitled: "IgA and IgG Antibody Responses Against Periodontitis-Associated Bacteria in Crohn's Disease", the authors present new results of the determination of IgA and IgG antibodies against Porphyromonas gingivalis, Tannerella forsythia, Aggregatibacter actinomycetemcomitans and Prevotella intermedia in the saliva of patients with Crohn's disease. The results show significant differences in the studied parameters between healthy individuals and Crohn's disease patients. The manuscript is well written. All results are described and the discussion is appropriate to the results obtained. This work demonstrates salivary Ig antibody responses against periodontitis-associated bacteria, but a deeper understanding requires further studies. I suggest to better elaborate conclusion to include prospect of this work.

Author Response

In their paper entitled: "IgA and IgG Antibody Responses Against Periodontitis-Associated Bacteria in Crohn's Disease", the authors present new results of the determination of IgA and IgG antibodies against Porphyromonas gingivalis, Tannerella forsythia, Aggregatibacter actinomycetemcomitans and Prevotella intermedia in the saliva of patients with Crohn's disease. The results show significant differences in the studied parameters between healthy individuals and Crohn's disease patients. The manuscript is well written. All results are described and the discussion is appropriate to the results obtained. This work demonstrates salivary Ig antibody responses against periodontitis-associated bacteria, but a deeper understanding requires further studies. I suggest to better elaborate conclusion to include prospect of this work.

Authors’ response: Authors thank to the reviewer for the constructive comments. Conclusions are rephrased as suggested (page 1 lines 29-30, page 5 lines184-186)

Round 2

Reviewer 1 Report

Regarding point 1, the sentence added in page 4, lines 142-146 is a result, therefore it should be moved from Discussion to Results section.

Author Response

Reveiwer 1:

Regarding point 1, the sentence added in page 4, lines 142-146 is a result, therefore it should be moved from Discussion to Results section.

Authors’ response: Authors once again thank to the reviewer for the constructive comment. The sentence is moved to the results section (Page 2 lines 68-71).